## [Editor Report · Decision Letter 0]

25 Jun 2021

PONE-D-21-11856

Ignoring the clustering effect changes some conclusions. Corrected analysis of: “Randomized nutrient bar supplementation improves exercise-associated changes in plasma metabolome in adolescents and adult family members at cardiometabolic risk”

PLOS ONE

Dear Dr. Jamshidi-Naeini,

Thank you for submitting your manuscript to PLOS ONE. After careful consideration, we feel that it has merit but does not fully meet PLOS ONE’s publication criteria as it currently stands. Therefore, we invite you to submit a revised version of the manuscript that addresses the points raised during the review process.

We look forward to receiving your revised manuscript.

Kind regards,

Eugene Demidenko, Ph.D.

Academic Editor

PLOS ONE

Additional Editor Comments:

Before finding the reviewers I read your manuscript myself and came to conclusion that (a) results of the reanalysis are similar to the original ones, (b) more technical information is required to evaluate your contribution.

Specifically,

1. The most important results of statistical analysis are the coefficients of the model; the p-values have secondary importance. Therefore I suggest reporting the coefficients in Table 1, along with the p-values.

2. Although an overseen clustering may produce a lower level of statistical significance (larger standard errors and p-values) it yields unbiased coefficients (Demidenko, E. Mixed Models. Theory and Applications with R, Hoboken, NJ: Wiley, 2013). Moreover, while the original paper uses GEE you employ LMM. Consequently, the results on p-values must be different. Indeed, as follows from Table 1 the proportion of the changed p-values (rows) is only about 5:30. In spite the correct claim of the effect of ignored clustering the results look pale.

3. Some technical description of the statistical model (dependent, independent variables, equations) and how exactly the clustering has been implemented is missing. Therefore it is impossible to evaluate the improvements and your contribution.

Despite strong criticism and weak results I still invite you to resubmit but with no guarantee that your paper will be published.

"In past 12 months, Dr. Brown received grants through his institution from Alliance for Potato Research & Education, Egg Nutrition Center, National Cattlemen’s Beef Association, NIH/NHLBI, and NIH/NIDDK. He has been involved in research for which his institution or colleagues have received grants or contracts from Center for Open Science, Gordon and Betty Moore Foundation, Hass Avocado Board, Indiana CTSI, National Cattlemen’s Beef Association, NIH/NHLBI, NIH/NIA, and Sloan Foundation. His wife is employed by Reckitt Benckiser. In the last thirty-six months, Dr. Allison has received personal payments or promises for same from: American Society for Nutrition; Alkermes, Inc.; American Statistical Association; Big Sky Health, Inc.; Biofortis; California Walnut Commission; Clark Hill PLC; Columbia University; Fish & Richardson, P.C.; Frontiers Publishing; Gelesis; Henry Stewart Talks; IKEA; Indiana University; Arnold Ventures (formerly the Laura and John Arnold Foundation); Johns Hopkins University; Kaleido Biosciences; Law Offices of Ronald Marron; MD Anderson Cancer Center; Medical College of Wisconsin; National Institutes of Health (NIH); Medpace; National Academies of Science; Sage Publishing; The Obesity Society; Sports Research Corp.; The Elements Agency, LLC; Tomasik, Kotin & Kasserman LLC; University of Alabama at Birmingham; University of Miami; Nestle; WW (formerly Weight Watchers International, LLC). Donations to a foundation have been made on his behalf by the Northarvest Bean Growers Association. Dr. Allison was previously an unpaid member of the International Life Sciences Institute North America Board of Trustees.

Dr. Allison’s institution, Indiana University, and the Indiana University Foundation have received funds or donations to support his research or educational activities from: NIH; USDA; Soleno Therapeutics; Dynamic AQS; American Egg Board; California Walnut Commission, Almond Board; Peanut Institute; Mondelez; National Cattlemen’s Beef Association; Eli Lilly and Co.; Reckitt Benckiser Group PLC; Alliance for Potato Research and Education; Whistle Labs, Inc;  American Federation for Aging Research; Dairy Management Inc; Arnold Ventures; the Gordon and Betty Moore Foundation; the Alfred P. Sloan Foundation; and numerous other for-profit and non-profit organizations to support the work of the School of Public Health and the university more broadly. Other authors report no disclosures. "

---

## [Author Response · Author response to Decision Letter 0]

16 Nov 2021

Dear Dr. Demidenko,

We appreciate your insightful comments on our manuscript. Our replies are in red below. 

We noted a theme of incertitude about our scientific contribution and the scope of the current re-analysis activity. Thus, we provide some clarification herein and in the revised manuscript (the version with track changes) (lines 36-40). 

Current manuscript is a re-analysis of an existing published study (i.e., referred to in the manuscript as ‘The Article’). Therefore, details regarding the significance, procedures, conduct, and conclusions of the study are provided in The Article. Our concern in the current manuscript is invalidity of the statistical analyses used for generating the published results. We present a valid statistical model and the corresponding results. We do not intend to address any research question or test any hypothesis beyond the results presented in The Article. Journals often handle this type of re-analysis as ‘correspondence’ so that it is linked to the previously published work. Because PLoS ONE editorial policy does not provide such an option, we were advised to submit our re-analysis as an original research article. Thus, our scientific contribution should be evaluated within the context of post-publication re-analysis and error correction. 

• The most important results of statistical analysis are the coefficients of the model; the p-values have secondary importance. Therefore, I suggest reporting the coefficients in Table 1, along with the p-values. 

• Although an overseen clustering may produce a lower level of statistical significance (larger standard errors and p-values) it yields unbiased coefficients (Demidenko, E. Mixed Models. Theory and Applications with R, Hoboken, NJ: Wiley, 2013).

The well-known consequences of ignoring clustering and nesting effects in the regression models for multilevel data are underestimated standard errors (and p-values) and unbiased regression coefficients [1, 2]. However, debate about the relative value of using p-values or frequentist testing, and whether regression coefficients or the p-values are superior in importance to one another is beyond the scope of this current reanalysis activity. Rather, the authors and the journal have chosen to publish a paper that was predicated on frequentist testing and p-values. Therefore, the testing, calculation, use, and interpretation of p-values should be conducted correctly, but in The Article they were not.

We present the model coefficients in supplementary Table 1. 

• Moreover, while the original paper uses GEE you employ LMM. Consequently, the results on p-values must be different.

Thanks for this insightful comment. In lines 97-111 of the manuscript, we outline the reasons why LMM is the legitimate approach to correctly account for clustering and nesting effects given the study design by Dr. Mietus-Snyder et al. Because our purpose is to focus on accounting for vs. ignoring clustering and nesting, we updated our analyses and Table 1 to compare LMMs with and without clustering and nesting effects. Please see Table 1 and lines 97-111.

• Indeed, as follows from Table 1 the proportion of the changed p-values (rows) is only about 5:30. In spite the correct claim of the effect of ignored clustering the results look pale.

Rigor1 and reproducibility2 are foundations of science advancement, both being critical for verification of the results generated using the reported methods. The results reported by Mietus-Snyder et al. are not generated using valid statistical methods for the study design (our reasoning outlined in the manuscript and elsewhere by our group and others [3, 4]). Thus, regardless of the magnitude of difference that valid statistical tests would make, those results are not verifiable3 (please see lines 130-135 too). In this regard, we highlight the following matters:

i. Editors, journals, authors, and the scientific community overall should have a commitment to accuracy and integrity of the science [5], even if the results only change modestly as a result of doing correct analyses.

ii. Hearkening back to our argument about the choice of frequentist testing, correcting the analyses changes qualitative conclusions by the standard operating procedures and social norms of the use of frequentist testing and readers have a right to know this.

iii. Finally, for an astute reader, especially someone writing a blue-ribbon panel report that might utilize such a paper, or a meta-analyst, without seeing the critique and re-analysis results, they will correctly discern that the original results cannot be relied upon because the analyses are invalid. They might therefore be unable to use the study and its findings, even though the study and its findings (once correctly analyzed) do have value. That is, an unverified result is a result that cannot be utilized and relied upon fairly. By our verifying (and correcting) the results, we allow them to be used by people who correctly understand that the current analyses are not correct.

Additionally, and as a side note, the proportion of the changed p-values is essentially a function of what one takes as the denominator and the significance level. As mentioned in the manuscript, we describe this proportion as “of 13 statistically significant effects at the 0.05 level in LMM models where clustering and nesting effects were ignored, one became non-significant, and of the four statistically significant effects at the 0.002 level, one became non-significant”. 

1Scientific rigor: “the strict application of the scientific method to ensure unbiased and well-controlled experimental design, methodology, analysis, interpretation and reporting of results.” (https://grants.nih.gov/policy/reproducibility/index.htm).

2Reproducibility: “An article is designated as reproducible if the [independent investigator] succeeds in executing the code on the data provided and produces results matching those that the authors claim are reproducible” [6].

3Verifiability: a study is said to be verifiable, and to have been verified, when: (a) the study is reproducible, and the results have been reproduced (by the definition of reproducibility above); and (b) a determination is made that the methods used to generate the results reproduced are valid methods. 

• Some technical description of the statistical model (dependent, independent variables, equations) and how exactly the clustering has been implemented is missing. Therefore, it is impossible to evaluate the improvements and your contribution.

The technical description of our statistical model is presented in the manuscript lines 75-89. We have deposited our statistical code at https://doi.org/10.5281/zenodo.5366705. 

References

1. Maas CJM, Hox JJ. The influence of violations of assumptions on multilevel parameter estimates and their standard errors. Computational Statistics & Data Analysis. 2004;46(3):427-40. doi: https://doi.org/10.1016/j.csda.2003.08.006.

2. Ntani G, Inskip H, Osmond C, Coggon D. Consequences of ignoring clustering in linear regression. BMC Medical Research Methodology. 2021;21(1):139. doi: 10.1186/s12874-021-01333-7.

3. Brown AW, Li P, Bohan Brown MM, Kaiser KA, Keith SW, Oakes JM, et al. Best (but oft-forgotten) practices: designing, analyzing, and reporting cluster randomized controlled trials. The American Journal of Clinical Nutrition. 2015;102(2):241-8. doi: 10.3945/ajcn.114.105072.

4. Murray DM, Varnell SP, Blitstein JL. Design and Analysis of Group-Randomized Trials: A Review of Recent Methodological Developments. American Journal of Public Health. 2004;94(3):423-32. doi: 10.2105/AJPH.94.3.423.

5. ICMJE. Recommendations for the Conduct, Reporting, Editing, and Publication of Scholarly Work in Medical Journals 2019. Available from: http://www.icmje.org/icmje-recommendations.pdf.

6. Peng RD. Reproducible research and Biostatistics. Biostatistics. 2009;10(3):405-8. doi: 10.1093/biostatistics/kxp014.

---

## [Editor Report · Decision Letter 1]

13 Dec 2021

PONE-D-21-11856R1Ignoring the clustering effect changes some conclusions. Corrected analysis of: “Randomized nutrient bar supplementation improves exercise-associated changes in plasma metabolome in adolescents and adult family members at cardiometabolic risk”PLOS ONE

Dear Dr. Yasaman Jamshidi-Naeini,

Thank you for submitting your manuscript to PLOS ONE. After careful consideration, we feel that it has merit but does not fully meet PLOS ONE’s publication criteria as it currently stands. Therefore, we invite you to submit a revised version of the manuscript that addresses the points raised during the review process.

In general, I’m not satisfied with how the authors addressed my critique. Specifically,

It should be mentioned in the paper that according to the theory (Demidenko, 2013) accounting for family clustering should not affect the coefficients but increase the standard errors and reduce the p-values. The current study confirms this expectation.I suggested reporting the coefficients themselves as well, but they are missing in the current version, again.The p-values from GEE and LME are not compatible because the underlying statistical models are different. Thus, we must compare LMM v1 versus LMM v2 to understand the effect of family clustering. As follows, the difference in p-values is negligible.Graphic presentation where x-axis is time would be very beneficial to see the differences between GEE, LMM v1, and LMM v2 models in addition to Table 1.Due to my previous point paper’s contribution is marginal. Yes, there is a slight improvement, but it is wrong to say that the analysis of “The Article” is incorrect. I suggest the language “adjustment” not saying that the previous analysis is wrong. This statement must be a part of the study limitations. Is it true that n=14 as follows from S1 Fig 1? Please submit your revised manuscript by Jan 27 2022 11:59PM. If you will need more time than this to complete your revisions, please reply to this message or contact the journal office at plosone@plos.org. Please include the following items when submitting your revised manuscript:A rebuttal letter that responds to each point raised by the academic editor and reviewer(s). You should upload this letter as a separate file labeled 'Response to Reviewers'.A marked-up copy of your manuscript that highlights changes made to the original version. You should upload this as a separate file labeled 'Revised Manuscript with Track Changes'.An unmarked version of your revised paper without tracked changes. You should upload this as a separate file labeled 'Manuscript'.

We look forward to receiving your revised manuscript.

Kind regards,

Eugene Demidenko, Ph.D.

Academic Editor

PLOS ONE

---

## [Author Response · Author response to Decision Letter 1]

14 Jan 2022

Dear Dr. Demidenko,

Thank you for your insightful comments, and giving us the opportunity to improve the manuscript. Our replies are in blue below.

1. It should be mentioned in the paper that according to the theory (Demidenko, 2013) accounting for family clustering should not affect the coefficients but increase the standard errors and reduce the p-values. The current study confirms this expectation.

2. I suggested reporting the coefficients themselves as well, but they are missing in the current version, again.

Thank you for your cogent comment. We agree that as stated in (Demidenko, 2013), [under certain assumptions, including that the cluster size is not correlated with cluster-specific treatment effects], ignoring within-cluster correlation yields unbiased estimates of regression coefficients ((1), pages 42 and 157), but artificially reduced (more significant) standard errors and p-values (2). Though, of course, they could change in any one sample. Similarly, there appears to be little bias (less than ±10%) in the estimates of regression coefficients across the methods that accommodate clustered data for 10 or fewer clusters with few (e.g., 7-14) observations per cluster (3, 4). 

We have now included coefficients in our manuscript per your suggestion (see Table 2 and lines 139-146). As we report in the manuscript, “regression coefficients of LMM v1 and LMM v2 are similar for fasting plasma ceramides and plasma sphingolipid bases, but slightly differ (3% or less) for amino acid metabolites.” However, because our purpose is to provide corrected statistical procedures within the context of methodologic choices of The Article (i.e., null hypothesis significance testing using p-values), we do not dwell extensively on the interpretation of regression coefficient estimates. 

3. The p-values from GEE and LME are not compatible because the underlying statistical models are different. Thus, we must compare LMM v1 versus LMM v2 to understand the effect of family clustering. As follows, the difference in p-values is negligible.

Thank you for these remarks. We agree that the comparison must be between the results derived from similar approach. Owing to your insightful comments on the previous version of our manuscript, we included LMM v1 and we are comparing LMM v2 vs. LMM v1 both in Table 1 and to summarize our findings. We also agree that the model that ignores clustering (LMM v1) yields roughly the same answers as the model that accounts for clustering and nesting (LMM v2). As we outline in the manuscript, we believe regardless of the magnitude of difference that valid statistical tests would make, results that are derived from inappropriate procedures for the design (which is the case for either GEE or LMM v1) are unsubstantiated and not verifiable. That is, incorrect analysis sometimes yields the correct answers and correct analysis sometimes yields incorrect answers (due to type I and/or type II errors). The fact that LMM v1 yields roughly the same answer as LMM v2 does not mean that ignoring clustering (and nesting) is a negligible issue. Our reanalysis of this article now allows the scientific community to interpret the results in the context of an analysis strategy appropriate to the study design, increasing the confidence in its findings.

4. Graphic presentation where x-axis is time would be very beneficial to see the differences between GEE, LMM v1, and LMM v2 models in addition to Table 1.

Thank you for this comment. We emphasize that our manuscript is not intended to be a methods paper. We are making a critique of an existing published paper because the results of that paper are produced using inappropriate procedures for its design. While we argue that the results are unsubstantiated, we cannot show that they are wrong or right because doing so would require knowing the ‘true’ answer which would be possible by conducting simulations or mathematical derivations to show the degrees of robustness. We propose that keeping the values in our table format is optimal to present our reanalysis, as the original authors did in their paper.

5. Due to my previous point paper’s contribution is marginal. Yes, there is a slight improvement, but it is wrong to say that the analysis of “The Article” is incorrect. I suggest the language “adjustment” not saying that the previous analysis is wrong. This statement must be a part of the study limitations. Is it true that n=14 as follows from S1 Fig 1?

• Thank you for raising this point. We agree that our writing has not been as clear as it could have been. Our statements about the ‘invalid’ analysis in The Article could be interpreted in two ways: 1) that the procedures used to produce the findings were incorrect for the purported properties of the study design, 2) that the results produced by the procedures used are not the results which would be produced by proper procedures. We clarify that we mean the former, not the latter: when we say the analysis is either wrong or invalid, we mean the procedures used to produce the results were incorrect, and when we say results, we mean the findings produced by the procedures used. In this case, the findings from the incorrect procedures and proper procedures are similar. We now have clarified this in our writing.

• We also agree that it is wise to include a study limitations section. Thank you for that suggestion. We now have included the following study limitations section to the manuscript:

“Our reanalysis has some limitations. First, due to the reasons described in the methods section for switching from GEE to LMM approach, it was not possible to directly compare the findings of our reanalysis that did account for clustering and nesting with the approach used in The Article that ignored those effects. Rather, we needed to conduct the reanalysis using an approach that would allow for clustering and nesting to be accounted for (LMM), but ignore them first (LMM v1). We then compared LMM v1 with the proper analysis that accounted for clustering and nesting (LMM v2). The second limitation is related to the extent we can draw general conclusions about degrees of robustness. We did not conduct simulations or mathematical derivations to show the degrees of robustness on average. Thus, our results only show the degrees of relative robustness of the results and conclusions of this paper (i.e., The Article) when switching from an analysis with incorrect procedures to one with correct procedures. That is, we cannot make any judgement about how often such changes may or may not occur or how large the errors would be. Finally, we did not explore other techniques such as parametric bootstrap to explore how the results would be different compared to LMM. These limitations can be addressed in future research.”

• According to the information shared with us by the authors, 20 clusters (i.e., family units) were randomized, among which two families dropped out, and 18 clusters (n=36) were included in the analysis. 

1. Demidenko E. Mixed models: theory and applications with R: John Wiley & Sons; 2013.

2. Johnson JL, Kreidler SM, Catellier DJ, Murray DM, Muller KE, Glueck DH. Recommendations for choosing an analysis method that controls Type I error for unbalanced cluster sample designs with Gaussian outcomes. Statistics in medicine. 2015;34(27):3531-45.

3. McNeish D, Stapleton LM. Modeling Clustered Data with Very Few Clusters. Multivariate Behavioral Research. 2016;51(4):495-518.

4. Maas CJM, Hox JJ. The influence of violations of assumptions on multilevel parameter estimates and their standard errors. Computational Statistics & Data Analysis. 2004;46(3):427-40.

---

## [Decision Letter · Decision Letter 2]

5 Aug 2022

PONE-D-21-11856R2

Ignoring the clustering effect changes some conclusions. Corrected analysis of: “Randomized nutrient bar supplementation improves exercise-associated changes in plasma metabolome in adolescents and adult family members at cardiometabolic risk”

PLOS ONE

Dear Dr. Yasaman Jamshidi-Naeini,

Thank you for submitting your manuscript to PLOS ONE. After careful consideration, we feel that it has merit but does not fully meet PLOS ONE’s publication criteria as it currently stands. Therefore, we invite you to submit a revised version of the manuscript that addresses the points raised during the review process.

I agree with the reviewer that incorporation of the family cluster effect does not change major conclusions. Consequently, I suggest the authors retitle the paper as follows: "Ignoring the clustering effect does not change major conclusions..." I understand that the authors may disagree with my decision and submit the paper to another journal. As you know, PLOS ONE is among a few scientific journals that publish "negative results" to avoid publication bias. Although adjusting for random effect is the right way to analyze the data it just happened that for this particular study mixed model did not make considerable difference. I encourage the authors to address the comments of the reviewer and resubmit the paper.

We look forward to receiving your revised manuscript.

Kind regards,

Eugene Demidenko, Ph.D.

Academic Editor

PLOS ONE

Journal Requirements:

Reviewers' comments:

Reviewer's Responses to Questions

**Comments to the Author**

1. If the authors have adequately addressed your comments raised in a previous round of review and you feel that this manuscript is now acceptable for publication, you may indicate that here to bypass the “Comments to the Author” section, enter your conflict of interest statement in the “Confidential to Editor” section, and submit your "Accept" recommendation.

Reviewer #1: (No Response)

Reviewer #2: (No Response)

2. Is the manuscript technically sound, and do the data support the conclusions?

Reviewer #1: Partly

Reviewer #2: Yes

3. Has the statistical analysis been performed appropriately and rigorously? 

Reviewer #1: Yes

Reviewer #2: Yes

4. Have the authors made all data underlying the findings in their manuscript fully available?

Reviewer #1: Yes

Reviewer #2: Yes

5. Is the manuscript presented in an intelligible fashion and written in standard English?

Reviewer #1: Yes

Reviewer #2: Yes

6. Review Comments to the Author

Reviewer #1: This proposed corrected analysis of the previous publication “Randomized nutrient bar supplementation improves exercise-associated changes in plasma metabolome in adolescents and adult family members at cardiometabolic risk” by Jamshidi-Naeini et al suggest that application of a clustered analysis changes some conclusions of the original article.

Three main questions emerge for the authors to consider:

I could not find any mention of the conclusions that were changed, only small changes in a minority of p-values. It is stated in the final paragraph that “most conclusions in The Article remain unchanged”.

1) Can the authors explain exactly which study conclusions are changed by clustered analyses and if any, what the significance of these changes to the article’s interpretation?

If none, the title might more appropriately read “Incorporation of the clustering effect in group randomized trial does not change the main study conclusions”. It is interesting that the reference 5 provided as an example of use of this clustered approach from the Allison group (Golzarri-Arroyo et al, 2020) similarly did not change study conclusions, but stated that accurately in the title.

Both of these PLOS ONE studies the Allison group has reanalyzed incorporating the clustering effect (Mietus-Snyder et al and Short et al) involved minority adolescent populations, who would be at heightened risk for socioeconomic disadvantage. Unfortunately, an important issue relevant to health disparity that a statistical manipulation alone cannot address, is fragmentation of traditional family units and potential breakdown in the predictability of household factors.

2) Can the authors delineate which household factors are felt to be most critical in justification of the proposed clustered analysis? If dietary factors pertain, does clustered analysis reveal any difference in the diets that were reported to be uniformly poor for both adolescents and adults in the original study? If genetic or epigenetic factors are implicated, what effects do nonbiological family units or shared custody have on clustered analyses? These issues raise an important point of discussion that might be expanded upon by the authors to help explain why the application of clustered analysis – at least in this study and the study reanalyzed in reference (5) does not appear to change the main study conclusions reached without clustering.

There are two sets of changes reported upon in this Correction – the first set due to an error in the original time-by-group p-values reported that has apparently been acknowledged by the study authors. It is not apparent why those p-value corrections have not already been published as a Correction as this should be the responsibility of the original study team, once alerted to their error. The title of this Correction by Jamshidi-Naeini implies that it is the application of clustered analysis, not these time x group interaction p-value corrections that changes some conclusions. It does seem apparent that the corrected time by group p-values do not change study conclusions (with the exception of aromatic amino acids tryptophan and phenylalanine, the latter still showing a trend with p = 0.07). This is not however a central observation and does not impact the article’s interpretation.

It is the second set of p-value changes that are contingent on the newly applied clustered analysis that would still be germane to the Correction submitted by Jamshidi-Naeini et al. The authors support their claim that this approach overcomes an important methodological issue in obesity research with several recent reviews on this topic. Their point is well taken, but the frequent use the terms “correctly” and “appropriately” in describing the application of a clustered analysis seems unnecessarily pejorative, since as already noted, the reanalysis has not actually been demonstrated to change the study’s conclusions. It of course changed p-values, as would be expected with a decrease in the number of independent observations, but only a small number of p-values that were nominally significant become slightly insignificant. Trends remained trends and the strongest observations, on which the original study conclusions are based, remain very significant.

A 2019 paper cited (ref 6) for which DB Allison is also senior author on “The Need for Greater Rigor in Childhood Nutrition and Obesity Research” states what Jamshidi-Naeini et al were sure to observe in that (quoting from ref 6) “Clustering reduces power”… going on to say “there are instances in which misanalysis of clustered childhood obesity interventions has led to unsubstantiated conclusions.” This certainly is true, but the authors have not demonstrated that it is true in this instance – which circles back to the first two questions above. Which study conclusions have actually been altered by the clustered analysis? And if none, why not?

A full reading of the original paper reveals a detailed baseline principal components analysis to reduce the dimensionality of the complex lipidomic data set incorporated in baseline correlograms with traditional cardiometabolic risk factors and lipoprotein subspecies. There are also change correlation analyses that reveal numerous hypothesis-generating associations independent of pre-post metabolite p-values. Much of the discussion in the original study is rooted in the correlograms that summarize this body of data.

3) Would clustered analysis lend any further insight to the principal component and correlation analyses? Can the authors explain why their reanalysis does not address these two prominent elements of the original paper?

One small point: It is stated in the Abstract and Introduction that this was a “recent” study but the original report states that the trial was completed in the summer of 2011. Given the 9-year lag to publication, the metabolomic and lipidomic analyses performed, as stated in the original paper, on samples preserved at -70o F, were however surely more recent when mass spectroscopy technologies to explore these novel biomarkers became more widely available.

Reviewer #2: The current investigators pointed out a possible design flaw in the manuscript. That is to say “In a published randomized controlled trial, household units were randomized to a nutrient bar supplementation group or a control condition, but the non-independence of observations within the same household (i.e., the clustering effect) was not accounted for in the statistical analyses." Assuming this to be the case, the investigators provided a side by side comparison of results (clustering vs. non clustering) and proceeded accordingly. The results are very similar. However, that would have to be debated with the original investigators and not this reviewer. The work was thorough and certainly appeared to be presented reasonably in a descriptive, and not methodologic, fashion as was pointed out by the investigators.

7. PLOS authors have the option to publish the peer review history of their article (what does this mean?). If published, this will include your full peer review and any attached files.

Reviewer #1: No

Reviewer #2: No

---

## [Author Response · Author response to Decision Letter 2]

19 Aug 2022

Dear Dr. Demidenko,

Thank you for your insightful comments, and giving us the opportunity to improve the manuscript. Our replies are in blue below.

Response to the Editor:

I agree with the reviewer that incorporation of the family cluster effect does not change major conclusions. Consequently, I suggest the authors retitle the paper as follows: "Ignoring the clustering effect does not change major conclusions..." I understand that the authors may disagree with my decision and submit the paper to another journal. As you know, PLOS ONE is among a few scientific journals that publish "negative results" to avoid publication bias. Although adjusting for random effect is the right way to analyze the data it just happened that for this particular study mixed model did not make considerable difference. I encourage the authors to address the comments of the reviewer and resubmit the paper.

Response: Thank you for these comments. We note that the fact that “[…] it just happened that for this particular study mixed model did not make considerable difference” is only known after we have reanalyzed the data using correct statistical procedures. That is, the results and conclusions published by Mietus-Snyder et al. are unsubstantiated because they are drawn from incorrect statistical methods. As a result of our reanalysis, results and conclusions derived from the original GEE models are verified or corrected. We have addressed the point about the title in our responses to reviewer 1.

Response to Reviewer 1:

1. I could not find any mention of the conclusions that were changed, only small changes in a minority of p-values. It is stated in the final paragraph that “most conclusions in The Article remain unchanged”. Can the authors explain exactly which study conclusions are changed by clustered analyses and if any, what the significance of these changes to the article’s interpretation? If none, the title might more appropriately read “Incorporation of the clustering effect in group randomized trial does not change the main study conclusions”. It is interesting that the reference 5 provided as an example of use of this clustered approach from the Allison group (Golzarri-Arroyo et al, 2020) similarly did not change study conclusions, but stated that accurately in the title.

Response: Thank you for raising the point about the title. We agree that a change in the title would present the consequences of accounting for clustering and nesting effects in this particular sample more clearly. We changed the title to “Accounting for the clustering and nesting effects verifies majority of conclusions. Corrected analysis of: “Randomized nutrient bar …””. The suggested title by the reviewer “Ignoring the clustering effect does not change major conclusions..." implies that we are encouraging researchers to ignore clustering (and nesting) in cluster randomized trials, which is not our intended message. 

We state in our manuscript that most conclusions remain unchanged when the comparison is between linear mixed effects models (LMM) that account for clustering and nesting and LMMs that ignore clustering and nesting. We precisely state variables for which p-values are changed from statistically significant in the incorrect analysis to insignificant when clustering and nesting are accounted for (Threonine and Arginine Bioavailability Ratio). The changes we made to the title will further address this matter. On the other hand, addressing the consequence of accounting for clustering and nesting on the article’s conclusions and interpretations should occur under the assumption that the article’s conclusions and interpretations are based on reproducible results, but they are not. Such comparison would require re-writing the article’s interpretations based on a) “corrected GEE” models and b) consistent specification of the significance level (i.e., 0.002 or 0.05). This is, as correctly mentioned by the reviewer, a “responsibility of the original study team”.

• Both of these PLOS ONE studies the Allison group has reanalyzed incorporating the clustering effect (Mietus-Snyder et al and Short et al) involved minority adolescent populations, who would be at heightened risk for socioeconomic disadvantage. Unfortunately, an important issue relevant to health disparity that a statistical manipulation alone cannot address, is fragmentation of traditional family units and potential breakdown in the predictability of household factors.

Response: Thank you for raising this important matter. We agree that socioeconomic disadvantage and health disparities among adolescences have critical importance. We agree that our reanalysis efforts do not address health disparity issues among adolescents. We do not intend to address any research question with that regard in our manuscript. In this manuscript we are making a critique of an existing published paper because the results of that paper are produced using inappropriate procedures for its design. 

2. Can the authors delineate which household factors are felt to be most critical in justification of the proposed clustered analysis? If dietary factors pertain, does clustered analysis reveal any difference in the diets that were reported to be uniformly poor for both adolescents and adults in the original study? If genetic or epigenetic factors are implicated, what effects do nonbiological family units or shared custody have on clustered analyses? These issues raise an important point of discussion that might be expanded upon by the authors to help explain why the application of clustered analysis – at least in this study and the study reanalyzed in reference (5) does not appear to change the main study conclusions reached without clustering.

Response: Thank you for raising these interesting matters. These questions present the potential to expand current knowledge and understanding of the factors that may explain correlation of observations within household units in cluster randomized trials. From a methods aspect, the effect of ignoring clustering would be more moderate in designs with more clusters and fewer individual participants per cluster compared to designs with fewer clusters and larger number of individuals per cluster. Situations in which families are randomly assigned to treatments are examples of smaller cluster sizes, such as the article under question here and the study we referred to in Golzarri-Arroyo et al. Any subjective explanation about dietary or genetic factors and their potential role in similarities or dissimilarities within or among households in this particular sample would be speculation, and not addressable within the context of our reanalysis.

• There are two sets of changes reported upon in this Correction – the first set due to an error in the original time-by-group p-values reported that has apparently been acknowledged by the study authors. It is not apparent why those p-value corrections have not already been published as a Correction as this should be the responsibility of the original study team, once alerted to their error. The title of this Correction by Jamshidi-Naeini implies that it is the application of clustered analysis, not these time x group interaction p-value corrections that changes some conclusions. It does seem apparent that the corrected time by group p-values do not change study conclusions (with the exception of aromatic amino acids tryptophan and phenylalanine, the latter still showing a trend with p = 0.07). This is not however a central observation and does not impact the article’s interpretation. 

It is the second set of p-value changes that are contingent on the newly applied clustered analysis that would still be germane to the Correction submitted by Jamshidi-Naeini et al. The authors support their claim that this approach overcomes an important methodological issue in obesity research with several recent reviews on this topic. Their point is well taken, but the frequent use the terms “correctly” and “appropriately” in describing the application of a clustered analysis seems unnecessarily pejorative, since as already noted, the reanalysis has not actually been demonstrated to change the study’s conclusions. It of course changed p-values, as would be expected with a decrease in the number of independent observations, but only a small number of p-values that were nominally significant become slightly insignificant. Trends remained trends and the strongest observations, on which the original study conclusions are based, remain very significant.

Response: Thank you for your cogent comment. 1) We agree that it is the responsibility of the original authors and the journal to correct the irreproducibility issue of the article. Both have been notified that a portion of the analyses in the article are not reproducible. We cannot comment on the reason why these errors have not been publicly acknowledged yet. 2) Our altered title should address the issue raised about the title. 3) The purpose of our manuscript is making a critique of an existing published paper because the results of that paper are produced using inappropriate procedures for its design. We have been factual throughout: majority of conclusions that are derived from correct procedures are the same as those derived from incorrect procedures. 4) Until appropriate corrections are made to the article, any comparison of our results with the article’s interpretations would require specification of which model (LMM or GEE) one is comparing our results with and which significance level (0.05 or 0.002) they are using. 5) We understand that referring to our analysis as a ‘correct’ or ‘appropriate’ analysis may be disapproved by some. We thus modified the language where our message was still clear without those words.

• A 2019 paper cited (ref 6) for which DB Allison is also senior author on “The Need for Greater Rigor in Childhood Nutrition and Obesity Research” states what Jamshidi-Naeini et al were sure to observe in that (quoting from ref 6) “Clustering reduces power”… going on to say “there are instances in which misanalysis of clustered childhood obesity interventions has led to unsubstantiated conclusions.” This certainly is true, but the authors have not demonstrated that it is true in this instance – which circles back to the first two questions above. Which study conclusions have actually been altered by the clustered analysis? And if none, why not?

Response: Thank you for these remarks. There should be a distinction between “unsubstantiated” results vs. “incorrect” results. There are two approaches to consider this matter: 1) that the procedures used to produce the findings were incorrect for the purported properties of the study design, 2) that the results produced by the procedures used are not the results which would be produced by proper procedures. We clarify that we mean the former, not the latter: when we say the analysis is either wrong or invalid, we mean the procedures used to produce the results were incorrect, and when we say results, we mean the findings produced by the procedures used. In this case, the findings from the incorrect procedures and proper procedures are similar.

Results and conclusions of the article are unsubstantiated (i.e., unsupported) without our reanalysis. That is, regardless of the magnitude of difference that valid statistical tests would make, results that are derived from inappropriate procedures for the design are unsubstantiated and not verifiable. Incorrect analysis sometimes yields the correct answers and correct analysis sometimes yields incorrect answers (type I and/or type II errors). While we argue that the results are unsubstantiated, we cannot show that they are wrong or right because doing so would require knowing the ‘true’ answer which would be possible by conducting simulations or mathematical derivations to show the degrees of robustness.

3. A full reading of the original paper reveals a detailed baseline principal components analysis to reduce the dimensionality of the complex lipidomic data set incorporated in baseline correlograms with traditional cardiometabolic risk factors and lipoprotein subspecies. There are also change correlation analyses that reveal numerous hypothesis-generating associations independent of pre-post metabolite p-values. Much of the discussion in the original study is rooted in the correlograms that summarize this body of data. Would clustered analysis lend any further insight to the principal component and correlation analyses? Can the authors explain why their reanalysis does not address these two prominent elements of the original paper?

Response: We did not intend to reanalyze every analysis in the article, and do not claim as such in our manuscript. Principal components analysis is out of the scope of our discussion in this manuscript. 

Response to Reviewer 2:

The current investigators pointed out a possible design flaw in the manuscript. That is to say “In a published randomized controlled trial, household units were randomized to a nutrient bar supplementation group or a control condition, but the non-independence of observations within the same household (i.e., the clustering effect) was not accounted for in the statistical analyses." Assuming this to be the case, the investigators provided a side by side comparison of results (clustering vs. non clustering) and proceeded accordingly. The results are very similar. However, that would have to be debated with the original investigators and not this reviewer. The work was thorough and certainly appeared to be presented reasonably in a descriptive, and not methodologic, fashion as was pointed out by the investigators.

Response: Thank you for reviewing our manuscript. We notified the original research team and the journal of the issues we detected.

---

## [Editor Report · Decision Letter 3]

13 Sep 2022

Accounting for clustering and nesting effects verifies most conclusions. Corrected analysis of: “Randomized nutrient bar supplementation improves exercise-associated changes in plasma metabolome in adolescents and adult family members [...]”

PONE-D-21-11856R3

Dear Dr. Yasaman Jamshidi-Naeini,

We’re pleased to inform you that your manuscript has been judged scientifically suitable for publication and will be formally accepted for publication once it meets all outstanding technical requirements.

Kind regards,

Eugene Demidenko, Ph.D.

Academic Editor

PLOS ONE

---

## [Editor Report · Acceptance letter]

19 Oct 2022

PONE-D-21-11856R3 

Accounting for clustering and nesting effects verifies most conclusions. Corrected analysis of: “Randomized nutrient bar supplementation improves exercise-associated changes in plasma metabolome in adolescents and adult family members [...]” 

Dear Dr. Jamshidi-Naeini:

I'm pleased to inform you that your manuscript has been deemed suitable for publication in PLOS ONE. Congratulations! Your manuscript is now with our production department. 

Kind regards, 

on behalf of

Dr. Eugene Demidenko 

Academic Editor

PLOS ONE